# Forced Overexpression and Knockout Analysis of SLC30A and SLC39A Family Genes Suggests Their Involvement in Establishing Resistance to Cisplatin in Human Cancer Cells

**DOI:** 10.3390/ijms252212049

**Published:** 2024-11-09

**Authors:** Margarita Kamynina, Julian M. Rozenberg, Artem S. Kushchenko, Sergey E. Dmitriev, Aleksander Modestov, Dmitry Kamashev, Nurshat Gaifullin, Nina Shaban, Maria Suntsova, Anna Emelianova, Anton A. Buzdin

**Affiliations:** 1World-Class Research Center “Digital Biodesign and Personalized Healthcare”, Sechenov First Moscow State Medical University, 119991 Moscow, Russia; margaret.kamynina@gmail.com (M.K.); modestov_a_a@staff.sechenov.ru (A.M.); kamashev_d_e@staff.sechenov.ru (D.K.); shaban_n_a@staff.sechenov.ru (N.S.); suntsova_m_v@staff.sechenov.ru (M.S.); emelyanova_a_a@staff.sechenov.ru (A.E.); buzdin_a_a@staff.sechenov.ru (A.A.B.); 2Endocrinology Research Center, Ulyanova St. 11, 117036 Moscow, Russia; 3Belozersky Institute of Physico-Chemical Biology, Lomonosov Moscow State University, 119234 Moscow, Russia; artkushchenko@gmail.com (A.S.K.); sergey.dmitriev@belozersky.msu.ru (S.E.D.); 4Shemyakin-Ovchinnikov Institute of Bioorganic Chemistry, 117997 Moscow, Russia; 5Faculty of Fundamental Medicine, Moscow State University, 119992 Moscow, Russia; info@fbm.msu.ru; 6PathoBiology Group, European Organization for Research and Treatment of Cancer (EORTC), 1200 Brussels, Belgium

**Keywords:** cancer, *SLC39A14*, *SLC30A3*, *SLC30A10*, *SLC39A8*, zinc, manganese, cisplatin, cetuximab, mitochondrial respiration, intracellular signaling pathways

## Abstract

**Abstract:** The metabolism of zinc and manganese plays a pivotal role in cancer progression by mediating cancer cell growth and metastasis. The SLC30A family proteins *SLC30A3* and *SLC30A10* mediate the efflux of zinc, manganese, and probably other transition element ions outside the cytoplasm to the extracellular space or into intracellular membrane compartments. The SLC39A family members *SLC39A8* and *SLC39A14* are their functional antagonists that transfer these ions into the cytoplasm. Recently, the *SLC30A10* gene was suggested as a promising methylation biomarker of colorectal cancer. Here, we investigated whether forced overexpression or inactivation of SLC30A and SLC39A family genes has an impact on the phenotype of cancer cells and their sensitivity to cancer therapeutics. In the human colon adenocarcinoma HCT-15 and duodenal adenocarcinoma HuTu80 cell lines, we generated clones with knockouts of the *SLC39A8* and *SLC39A14* genes and forced overexpression of the *SLC30A3*, *SLC30A10,* and *SLC39A8* genes. Gene expression in the mutant and control cells was assessed by RNA sequencing. The cell growth rate, mitochondrial activity, zinc accumulation, and sensitivity to the drugs cetuximab and cisplatin were investigated in functional tests. Overexpression or depletion of SLC30A or SLC39A family genes resulted in the deep reshaping of intracellular signaling and provoked hyperactivation of mitochondrial respiration. Variation in the expression of the SLC30A/SLC39A genes did not increase the sensitivity to cetuximab but significantly altered the sensitivity to cisplatin: overexpression of *SLC30A10* resulted in an ~2.7–4 times increased IC50 of cisplatin, and overexpression of *SLC30A3* resulted in an ~3.3 times decreased IC50 of cisplatin. The SLC30A/SLC39A genes should be considered as potential cancer drug resistance biomarkers and putative therapeutic targets.

## 1. Introduction

The solute carrier (SLC) superfamily is the second largest family of membrane proteins in the human genome, which is responsible for the transport of a wide range of substrates, such as nutrients, organic ions, and inorganic ions. Members of this superfamily are found in all cell membranes and intracellular membranes of organelles, with the exception of the nuclear membrane [1]. SLC proteins control key physiological functions and may function as both tumor suppressors [2,3] and oncogenes [4]. For example, the tumor suppressive function of SLC may be related to histone deacetylase (HDAC) inhibition and intracellular pH regulation [5]. In recent years, the role of the SLC family in carcinogenesis has received increasing attention. Some of its members are meaningfully activated in tumor cells, apparently due to higher energy and nutritional requirements [6].

The SLC30 family consists of 10 members. They are involved in the transport of zinc (Zn) and manganese (Mn), which are so important to counteract programmed cell death, as Zn has been shown to play an important role in many cancers [5]. In addition, Mn is an essential metal that is required for various cellular enzymatic activities [7,8]. It can also inhibit acetylation of histones H3 and H4 by increasing and decreasing HDAC and histone acetyltransferase (HAT) activities, respectively, which ultimately damages chromatin structure and induces apoptosis [9]. However, the exact mechanisms of the Zn and Mn involvement in the progression, invasion, growth, metastasis, and tumor response to treatment are still largely unexplored [10].

The molecular function of the *SLC30A10*—ZnT10 protein is to pump Zn and Mn ions out of the cell; it is also involved in preventing manganese-induced cell death and provides zinc transport to early endosomes and endosome recycling to prevent zinc toxicity [11,12,13,14,15]. Based on bioinformatics analysis of microarray expression data from the GEO database, *SLC30A10* was previously identified as one of the top ten candidate genes marking colorectal cancer [16].

RNA sequencing data showed that *SLC30A10* expression in colorectal cancer negatively correlates with its functional antagonists *SLC39A14* and *SLC39A8* [10]. Their products, in turn, mediate cellular uptake of divalent ions of manganese, zinc, iron and cadmium [17,18].

Of note, a known paralog of *SLC30A10*, the *SLC30A1* gene, which also mediates the efflux of Zn and possibly calcium ions but not Mn [19], is oppositely regulated [10].

Another important function of *SLC30A10* is the zinc transporter-dependent regulation of the signal transduction pathway through the EGFR/ERK regulatory pathway in endosomes by heterodimerization with another zinc transporter protein *SLC30A3* [20,21,22]. Importantly, the expression profile of *SLC30A3* in colorectal cancer (downregulated in 92% of samples) correlates with *SLC30A10* [10]. The mechanism of *SLC30A3* suppression in colorectal cancer is currently poorly understood. Unlike the *SLC30A10* gene, which is methylated in colorectal cancer in the majority of cases [23,24], according to the COSMIC knowledge base [“https://cancer.sanger.ac.uk/cosmic (accessed on 31 May 2024)”], it can be explained by methylation in only ~7% of cases, which likely implies that other, non-methylation-related mechanisms of suppression are also involved.

Nevertheless, the data available through the COSMIC browser show that for a small group of colorectal cancer patients (~6%), there is increased expression of *SLC30A10*. It may also be associated with *SLC30A3* activation (~1.5% of cases in the COSMIC data). The mechanism of *SLC30A10* modulation associated with carcinogenesis may therefore be more complex than just the suppression of the activity of this gene. For example, it may be related to Mn transporter function as well as to the dimerization of *SLC30A3*/*SLC30A10* proteins that affect the EGFR/ERK signaling axis.

Interestingly, genes forming heterodimers of *SLC30A3* and *SLC30A10* proteins have relatively high normalized mutation rates and belong to the top 20–25% of mutated genes [25]. Other SLC genes mentioned in the context of this study, i.e., *SLC39A14* and *SLC39A8*, have 2–5 times lower mutation frequencies [26].

Zn²⁺ and Mn²⁺ can catalyze the conversion of superoxide to H₂O₂ through superoxide dismutase, specifically Cu/Zn-SOD1 in the cytoplasm and Mn-SOD2 in mitochondria, serving as a significant detoxification pathway within cells [10]. Mitochondrial respiration plays a pivotal role in cancer progression, primarily through the overproduction of reactive oxygen species (ROS) in cancer cells. This overproduction not only contributes to genomic instability but also alters gene expression and participates in critical signaling pathways that promote tumor development [27]. Mitochondria are essential metabolic organelles that produce cellular energy through oxidative phosphorylation (OXPHOS) while simultaneously generating ROS as a by-product [28]. They also regulate apoptosis through mechanisms involving the mitochondrial permeability transition pore (mtPTP) [29]. Excessive amounts of ROS significantly impact the enzymes of the respiratory chain and compromise or disrupt the mitochondrial antioxidant system, creating an environment conducive to cancer progression [30,31]. Moreover, oxidative damage induced by ROS can lead to mutations in both mitochondrial and nuclear DNA, thus impairing the oxidative phosphorylation mechanisms [32]. This impairment results in further ROS production, establishing a “vicious cycle” where mitochondrial dysfunction, genomic instability, and cancer development are interlinked. This cycle is detrimental, as increased ROS levels can alter DNA methylation and histone modifications in cancer cells [33,34]. Furthermore, recent studies have shown that treatment with targeted tyrosine kinase inhibitors such as lapatinib can upregulate proteins associated with mitochondrial function and metabolic pathways linked to the tricarboxylic acid (TCA) cycle [35].

Here, we for the first time investigated whether forced overexpression or inactivation of SLC30A and SLC39A family genes has an impact on the phenotype of cancer cells and their sensitivity to cancer therapeutics. In the human colon adenocarcinoma HCT-15 and duodenal adenocarcinoma HuTu80 cell lines, we generated clones with knockouts of the *SLC39A8* and *SLC39A14* genes and forced overexpression of the *SLC30A3*, *SLC30A10*, and *SLC39A8* genes. Gene expression in the mutant and control cells was assessed by RNA sequencing. The cell growth rate, mitochondrial activity, zinc accumulation, and sensitivity to the drugs cetuximab and cisplatin were investigated in functional tests. We found that the overexpression or depletion of SLC30A or SLC39A family genes resulted in the deep reshaping of intracellular signaling and provoked the hyperactivation of mitochondrial respiration. Variations in the expression levels of the SLC30A/SLC39A genes did not increase the sensitivity to cetuximab but significantly altered the sensitivity to cisplatin: overexpression of *SLC30A10* resulted in an ~2.7–4 times increased IC50 of cisplatin, and overexpression of *SLC30A3* resulted in an ~3.3 times decreased IC50 of cisplatin. These results suggest that the SLC30A/SLC39A genes should be considered as potential cancer drug resistance biomarkers and putative therapeutic targets.

## 2. Results

### 2.1. Construction and Basic Characterization of SLC39A8 and SLC39A14 Knockout Cell Lines

*SLC39A8*- and *SLC39A14*-mutant cells were generated by CRISPR/Cas9 gene editing. For the *SLC39A8* gene, exon 3 was specifically targeted, which resulted in 17-base-pair and 20-base-pair deletions inactivating both alleles of the *SLC39A8* gene in an isogenic clone of HuTu80 (Figure 1A). In HCT-15 cells, no *SLC39A8* knockout cell line could be obtained. For the *SLC39A14* gene, exon 2 was specifically targeted, which resulted in 2-base-pair and 7-base-pair deletions inactivating both alleles in an isogenic clone of HuTu80 (Figure 1B) and in a 2-base-pair deletion and 1-base-pair insertion inactivating both alleles in an isogenic clone of HCT-15 (Figure 1C).

Growth curve analysis demonstrated comparable growth rates between the parental and knockout HuTu80 and HCT-15 cell lines with similar doubling times (Figure 2A–C). The cell population doubling time for HuTu80 WT cells was 1.3 days, while the *SLC39A8* and *SLC39A14* knockout variants exhibited doubling times of 1.5 and 1.2 days, respectively (Figure 2A,B). In HCT-15 WT cells, the doubling time was 1.1 days, whereas the *SLC39A14* knockout resulted in a doubling time of 1.3 days (Figure 2C). In HuTu80 GFP cells, the doubling time was 1.5 days, whereas the cells with overexpression of the *SLC30A3*, *SLC30A10*, and *SLC39A8* genes showed doubling times of 1.3, 1.3, and 1.7 days, respectively (Figure 2D–F). The doubling time for HCT-15 GFP cells was 1.1 days, while the cells with overexpression of the *SLC30A3* and *SLC30A10* genes exhibited 1.2- and 1.5-day doubling times, respectively (Figure 2G,H). Thus, HCT-15 cells with *SLC30A10* overexpression showed a mild decrease in proliferation compared to all other HCT-15 cell lines, whereas no difference was visible for the HuTu80 cells. The distribution of cell cycle phases was not affected in the knockout cells compared to the WT controls (Figure 3E,F).

RNA sequencing assay revealed an ~60-fold decreased expression level of *SLC39A8* in the respective knockout HuTu80 cell line compared to wild-type cells (Figure 4a), as well as ~5.7 and 6-fold decreased expression of *SLC39A14*, respectively, in the established HuTu80 and HCT-15 knockout cell lines for the *SLC39A14* gene (Figure 4a,b).

We also assessed the effects of these gene knockouts on cellular mitochondrial respiration using the resazurin assay (Figure 3C,D). This assay, which is used to measure the conversion of resazurin to fluorescent resorufin by NAD(P)H, serves as a reliable indicator of mitochondrial activity in mammalian cells. Compared to the wild-type cells, our results indicated 2.2-fold and 2.3-fold increases in the NAD(P)H concentration in *SLC39A14* knockout HuTu80 cells after 24 and 72 h of incubation, respectively. In HCT-15 cells, a 1.5-fold increase was detected for *SLC39A14* knockout cells after 72 h of incubation. For *SLC39A8* gene knockout, a 1.3-fold increase in the NAD(P)H concentration was observed in HuTu80 cells after 72 h (Figure 3C,D). Thus, the resazurin assay showed an enhanced aerobic respiration in cells with both types of SLC39A family gene knockouts.

### 2.2. Construction and Basic Characterization of Cell Lines Overexpressing SLC39A8, SLC30A3, and SLC30A10 Genes

To generate stable cell lines overexpressing the *SLC39A8*, *SLC30A3*, and *SLC30A10* genes, we used the pCDH-EF1-MCS-IRES-copGFP lentiviral vector where the GFP marker was substituted with a puromycin resistance selection marker, and the FLAG-tagged open reading frames of the SLC genes were incorporated into the vector to enable the selection of successfully transduced cells. After transduction, stable cell lines were selected using puromycin treatment. Western blot analysis with FLAG-specific antibodies and GAPDH used as a loading control demonstrated successful overexpression of *SLC39A8*, *SLC30A3*, and *SLC30A10* in the respective cell lines (Figure 5a,b). However, no *SLC39A8*-overexpressing clone of the HCT-15 cell line could be obtained.

In puromycin-resistant HCT-15 cells transduced with the *SLC30A3* and *SLC30A10* lentiviral supernatants, the molecular weights of bands were 44 kD and 55 kD, respectively (Figure 5a). The molecular weights of bands detected in the puromycin-resistant HuTu80 cells transduced by *SLC30A3*, *SLC30A10*, and *SLC39A8* lentiviral supernatants were 44 kD, 55 kD, and 52 kD, respectively (Figure 5b). There were no overexpressed proteins in the WT cells because these cells were not transduced with the lentiviral vector containing the FLAG-tagged open reading frames of the *SLC39A8*, *SLC30A3*, and *SLC30A10* genes. RNA sequencing analysis further revealed increased gene expression levels compared to the control cells transformed with GFP-FLAG vector. In particular, the transcript levels of the *SLC39A8*, *SLC30A3*, and *SLC30A10* genes in the HuTu80 cells transformed with the respective overexpression constructs were increased by ~1000-, 64-, and 776-fold, respectively (Figure 4a). In HCT-15 cells transformed with the respective overexpression constructs, the *SLC30A3* and *SLC30A10* transcript levels were increased relatively to GFP-transformed controls by ~2050 and 194 times, respectively (Figure 4b).

The population doubling time for the control GFP and *SLC30A3-*, *SLC30A10-,* and *SLC39A8*-overexpressing HuTu80 cells varied non-significantly from 1.3 to 1.7 days (Figure 2D–F). The same was true also for the control HCT-15 GFP cells and for the HCT-15 cells overexpressing the *SLC30A3* and *SLC30A10* genes, where the population doubling time varied in the range of 1.1–1.5 days (Figure 2G,H).

The NAD(P)H levels measured by the resazurin assay indicated an ~1.8-fold increase in mitochondrial respiration in HCT-15 cells overexpressing *SLC30A10* after 96 h of incubation (Figure 3A). Additionally, there was an ~1.5-fold increase in mitochondrial respiration activity in *SLC30A10*-overexpressing HuTu80 cells after 72 h and an ~1.3-fold increase after 96 h of incubation. Furthermore, *SLC39A8*-overexpressing HuTu80 cells exhibited an ~1.6-fold increase in mitochondrial respiration after 72 h and an ~1.3-fold increase after 96 h (Figure 3B). These results demonstrate that overexpression of both the *SLC39A8* and *SLC30A10* genes enhances aerobic respiration in the affected cell lines (Figure 3A,B).

### 2.3. Deletion and Overexpression of SLC Family Genes Change Cytoplasmic Zinc Homeostasis

To assess whether the expression changes introduced by knockout or overexpression of the genes *SLC39A8*, *SLC39A14*, *SLC30A3*, and *SLC30A10* could affect intracellular zinc concentrations, we used the vital fluorescent zinc-specific probe FluoZin-3 AM to measure free zinc levels in the experimental and control cell lines. These included wild-type (WT), knockout (KO; for the genes *SLC39A8* and *SLC39A14*), and overexpressing (OE; for the genes *SLC39A8*, *SLC30A3*, and *SLC30A10*) HuTu80 and HCT-15 cell lines.

In both cell lines with knockout of the *SLC39A14* gene and with overexpression of the *SLC30A10* gene and in HuTu80 cells with knockout of the *SLC39A8* gene, we observed a reduced intracellular zinc concentration compared to WT cells (Figure 6 and Figure 7). In contrast, both cell lines overexpressing the *SLC30A3* gene demonstrated a significant increase in intracellular zinc levels with a granular fluorescence pattern (Figure 6 and Figure 7). This suggests that *SLC30A3* facilitates the transport of zinc from the cytoplasm into specific intracellular membrane compartments, consistent with its known role in transporting zinc into synaptic vesicles in neurons [36]. Furthermore, HuTu80 cells overexpressing *SLC39A8* exhibited a marked increase in cytoplasmic zinc levels (Figure 6), which is also consistent with its known functional role of transferring zinc inside the cytoplasm [37].

In turn, knockout of the *SLC39A8* and *SLC39A14* genes, which function in the transfer of zinc ions into the cytoplasm [38], results in the decreased fluorescence of the cells. Taken together, these findings support the known roles of the *SLC30A3*, *SLC30A10*, *SLC39A8*, and *SLC39A14* genes in intracellular zinc metabolism. These functional tests also confirm the experimental models used in this study.

### 2.4. Altered Expression of SLC30A and SLC39A Family Genes Influences Cancer Cell Sensitivity to Chemotherapeutic Drugs

Cancer cell resistance to chemotherapy is a well-known therapeutic obstacle in medical oncology. In this study, we intended to investigate whether the altered expression of the major zinc and manganese transfer genes has an effect on the sensitivity to cancer drugs. First of all, we explored the effects of knockout or overexpression of the SLC30A and SLC39A family genes on the cytotoxic activities of the *EGFR*-targeted therapeutic antibody cetuximab [39] and of the DNA-damaging agent cisplatin. The latter is a platinum-containing drug that is used as the primary component of multiple treatment schemes in various cancer types [40,41,42], including colorectal cancer (CRC). In *KRAS*-wild-type CRC, cetuximab is the therapeutic of choice that has shown high efficacy and has a significant positive impact on patient survival [43]. However, when activating hotspot *KRAS* mutation is present, cetuximab is inefficient, as the downstream proliferative signaling is aberrantly activated [44].

One of our model cell lines (HCT-15) was derived from colorectal cancer and harbors the *KRAS* mutation in addition to mutations in the *APC*, *B2M*, *CHEK2*, *BRCA2*, *PIK3CA*, and *TP53* genes [“https://www.cellosaurus.org/CVCL_0292 (accessed on 21 December 2021)”]. Thus, theoretically, it should be insensitive to cetuximab and highly sensitive to platinum drugs, as inactivating *BRCA2* and *CHEK2* mutations cause DNA repair deficiency and increase the efficacy of platinum DNA-damaging therapeutics [45]. The second model cell line (HuTu80) has a mutation in the beta-catenin *CTNNB1* gene, which is also connected with poor sensitivity to cetuximab [“https://www.cellosaurus.org/CVCL_1301 (accessed on 21 December 2021)”].

In a recent study, the *SLC30A10* gene was identified as the most informative biomarker for the detection and screening of CRC, exhibiting methylation in the majority of the analyzed CRCs [24]. We therefore investigated whether altered expression of this gene, of its functional homolog *SLC30A3*, and of their functional antagonists the SLC39A family member genes *SLC39A8* and *SLC39A14* may be related to altered sensitivity to CRC drugs.

In all the knockout or overexpression mutant cell lines, we found no statistically significant difference in the sensitivity to cetuximab, where the half-inhibitory concentration (IC50) of cetuximab remained very high in all the experiments (>300 nM, versus 0.3–2 nM for the susceptible cell lines [46].

On the other hand, the treatment with cisplatin showed a different pattern. In HuTu80 cells, the IC50 for cisplatin was ~2.7 times higher in the cells with overexpression of *SLC30A10* than in the GFP-overexpressing controls (Figure 8D, Table 1). In HCT-15 cells, the IC50 of cisplatin was increased even more: ~4 times (Figure 8A, Table 1). This clearly indicates chemotherapy resistance associated with the increased expression of *SLC30A10*.

In contrast, overexpression of *SLC30A3* in HuTu80 resulted in an ~3.3 times lower IC50 of cisplatin, which suggests greater cancer cell sensitivity to chemotherapy associated with this gene expression (Figure 8E, Table 1).

This observed functional contrast between the two SLC30A family members may directly reflect their functional activities. These are the efflux of zinc and other transition elements from the cytoplasm towards the outside of the cell (*SLC30A10*; overexpression is associated with better survival of cancer cells on cisplatin) and the accumulation of these elements in the membrane structures of the cell (*SLC30A3*; overexpression is associated with poor survival of cancer cells on cisplatin).

Taken together, these findings may suggest that the expression and methylation levels of the genes *SLC30A10* and *SLC30A3* may have predictive value for the sensitivity of cancer cells to platinum drugs, also inthe case of *KRAS*-mutated CRC cells.

### 2.5. mRNA Profiling of HCT-15 and HuTu80 Cells with Knockout or Overexpressed SLC30A and SLC39A Family Genes

We analyzed gene expression levels by RNA sequencing for all the control, knockout, and overexpression cell lines obtained in this study. All the control cells were profiled in triplicate. The complete sequencing data are available at GSE276849—GEO Publications (nih.gov). The principal component analysis (PCA) of log-transformed gene expression levels revealed a strong cell type-specific (HuTu80 or HCT-15) clustering of the samples along the first component explaining 73.73% of the variance (Figure 9). The differential gene expression was established using the following criteria: FDR-adjusted *p*-value < 0.1; fold change > 2.

#### 2.5.1. Overexpression of *SLC30A3*

Overexpression of the *SLC30A3* gene resulted in 11/61 up/downregulated genes in the HCT-15 cell line and in 93/68 up/downregulated genes in the HuTu80 cell line, compared to the GFP-overexpressing controls (Appendix A). Among them, there was only one differential gene commonly regulated in both cell types—the upregulation of the *SLC30A3* gene itself. We also calculated the pathway activation level (PAL) values for 3044 intracellular molecular pathways using the OncoboxPD (Oncobox pathway databank) [47]. We found 26/127 and 89/125 significantly differentially up/downregulated pathways for the HCT-15 and HuTu80 cell types, respectively. Nine upregulated (among them: caspase cascade, cellular apoptosis, PI3K/Akt, GSK3, and adherens junction pathways) and twenty-nine downregulated (among them: KEGG pathways in cancer, RAS, mTOR, JAK-STAT, and DNA repair pathways) pathways were common for both cell types (Appendix A). Both intersections were non-random according to the permutation statistical test [48], *p* < 0.001

#### 2.5.2. Overexpression of *SLC30A10*

Overexpression of the *SLC30A10* gene resulted in 28/47 and 62/126 up/downregulated genes in HCT-15 and HuTu80 cells, respectively. Among the intersected genes, the only one upregulated gene was *SLC30A10* itself, and the only one downregulated gene was the *HIST1H4J* gene encoding histone H4 family member protein (Appendix A).

In the pathway analysis, we found 32/35 and 146/192 up/downregulated pathways for the HCT-15 and HuTu80 cell types, respectively (Appendix A). Among them, there were five overlapping upregulated pathways (including VEGFR2 signaling, NCI beta3 integrin cell surface interactions, and KEGG platelet activation pathways) and five downregulated pathways (including Akt signaling, ERK signaling, TGF/WNT regulation, and viral carcinogenesis pathways). However, in this case, both interactions did not pass the permutation test for randomness (*p*-value exceeded 0.05).

#### 2.5.3. Knockout of *SLC39A14*

In total, 60/150 and 153/180 differential up/downregulated genes were found in HCT-15 and HuTu80 cells, respectively (Appendix A), where five downregulated genes were common in both cell lines (permutation test *p* < 0.01): *SLC39A14* itself, *DHRS3*, *KCNN4*, *MB*, and *ZBTB20*.

On the molecular pathway level, there were 27/125 and 182/294 up/downregulated pathways in HCT-15 and HuTu80 cells, respectively (Appendix A), with three common upregulated pathways (permutation test for randomness not passed; pathways for heparan sulfate biosynthesis, cell adhesion molecules, and GAG biosynthesis) and 42 common downregulated pathways (permutation test *p* < 0.001). The latter group included Akt signaling, cAMP, ILK signaling, PTEN, NCI TNF receptor signaling, regulation of RhoA, cell adhesion regulated by cadherins, integrin cell surface interactions, epithelial-mesenchymal transition (EMT), and other pathways.

#### 2.5.4. Knockout and Overexpression of *SLC39A8*

In HuTu80 cells, we were able to obtain both knockout and overexpressing cell lines for the gene *SLC39A8*. We then compared genes and molecular pathways that were reciprocally regulated in the *SLC39A8* knockout and *SLC39A8*-overexpressing HuTu80 cells. We found 320/223 up/downregulated genes in *SLC39A8* knockout and 39/157 up/downregulated genes in *SLC39A8*-overexpressing HuTu80 cells (Appendix A). Among them, there were four common genes (*DRP2*, *FOXJ1*, *RGS6*, and *SLC39A8* itself) that were downregulated in knockout and upregulated in overexpressing cell lines (permutation test *p* < 0.001). Among them, the *FOXJ1* gene was previously reported as a CRC-promoting factor [49].

In addition, there were also 16 common genes between the *SLC39A8* knockout-upregulated and *SLC39A8*-overexpressing cell line-downregulated gene sets (permutation test *p* < 0.001, Appendix A); interestingly, among them were the *SLC30A10* gene, which evidences a coordinated opposite expression pattern of the SLC30A and SLC39A gene family members.

Furthermore, at the molecular pathway level (Appendix A), we found 89 pathways in common that were upregulated in *SLC39A8* knockout and downregulated in *SLC39A8*-overexpressing HuTu80 cells (*p* < 0.01). Among them were the pathways of the caspase cascade in apoptosis, TNF signaling, MAPK signaling, Ras signaling, ErbB signaling, Wnt signaling, TGF-beta, mTOR, chemotaxis driven by IL-8 and LTB4, cell adhesion regulated by cadherins, Notch signaling, STAT3, proteoglycans in cancer, development of immune synapse, HGF signaling, p38 signaling, and JNK signaling. These pathways represent critical nodes in the interactome network governing the fate of cancer cells.

Another set of 12 pathways that were common for the knockout-down- and overexpression-upregulated cells did not pass the randomness permutation test.

## 3. Discussion

In this study, we for the first time used genetically modified human cancer cell lines to profile the effects of abnormally regulated genes of the SLC39A and SLC30A families, which mediate the transport of zinc and manganese inside and outside the cytoplasm, respectively. Using two model human cancer cell lines (HCT-15 for colorectal cancer and HuTu80 for duodenal carcinoma) we generated HCT-15 and HuTu80 knockout cell lines for the *SLC39A14* gene and an HuTu80 knockout cell line for the *SLC39A8* gene. However, we were unable to generate *SLC39A8* gene knockout in HCT-15 cells or *SLC30A3* and *SLC30A10* gene knockouts in HCT-15 and HuTu80 cells due to the decreased viability of the transformed cells.

We also generated lentiviral-transformed HuTu80 and HCT-15 cell lines with stable overexpression of the *SLC30A3* and *SLC30A10* genes. In addition, we also generated a HuTu80-based cell line with stable overexpression of the *SLC39A8* gene. However, due to the decreased viability of the transformed cells, we could not generate cell lines stably overexpressing *SLC39A8* in HCT-15 cells or *SLC39A14* in both cell types.

The latter fact may suggest that forced over/underexpression of the SLC30A or SLC39A family genes may strongly influence cell viability. This presumption was further supported by our findings of the differential regulation of multiple core cancer cell survival, differentiation, and proliferation pathways in response to altered expression of the SLC30A or SLC39A genes.

This was also supported by our observation that alterations in the expression of SLC30A/SLC39A genes increased the activity of mitochondria, as evidenced by the increased levels of intracellular NAD(P)H. Mitochondrial respiration is integral to cellular energy production and redox balance, processes that can influence how cells respond to stress, including chemotherapy. Overexpression or knockout of SLC30A/SLC39A family genes may modulate intracellular ion homeostasis (particularly zinc and manganese), which could, in turn, affect mitochondrial function. These shifts in mitochondrial activity might alter the cell’s capacity to handle cisplatin-induced DNA damage and oxidative stress, thereby influencing drug sensitivity.

Although variation in the expression of the SLC30A/SLC39A genes did not increase the sensitivity of *KRAS*- and *CTNNB1*- mutated cell lines to cetuximab, it did significantly alter the sensitivity of cells to cisplatin. In these experiments, we found that the overexpression of *SLC30A10,* which functions in the removal from the cytoplasm towards the outer space of zinc, manganese, and probably other ions of transition elements, results in an ~2.7–4 times increase in the half-inhibitory concentration of cisplatin. Interestingly, overexpression of the gene *SLC30A3*—another SLC30A family member, resulted in the opposite effect: an ~3.3 times decreased IC50 of cisplatin. This discrepancy can be explained by the differences in the mechanisms of action of these two gene products. As it was found here and in previous works, *SLC30A3* removes zinc and other ions of transition elements from the cells, whereas *SLC30A10* stores them in the cell by transferring them from the cytoplasm to the membrane vesicles and possibly other membrane structures. Although the overexpression of *SLC39A8* in HuTu80 cells decreased the IC50 for cisplatin by ~1.5 times, this reduction, while measurable, is less than two-fold. Thus, although there is a modest increase in sensitivity to cisplatin, the change may not be robust enough to suggest that *SLC39A8* overexpression alone is a key determinant of cisplatin response in duodenal carcinoma cells. In contrast, knockout of *SLC39A8* did not significantly affect the sensitivity of HuTu80 cells to cisplatin, indicating that the basal expression of *SLC39A8* is not a critical factor in modulating cisplatin sensitivity. This could imply that *SLC39A8* plays a more nuanced role in cellular homeostasis, where overexpression may subtly impact zinc-mediated cellular processes, but knockout is compensated for by other pathways, preventing a noticeable change in drug sensitivity.

Thus, this SLC30A-mediated removal/storage interplay is most possibly implicated in the mechanisms of cellular sensitivity to platinum chemotherapy, although the mechanism of this phenomenon remains underexplored. As such, we speculate that the expression status of these SLC30A family members—and of their functional antagonists, the SLC39A genes—may be informative for the prediction of tumor responsiveness on platinum drugs. This hypothesis will be investigated in our further studies.

## 4. Materials and Methods

The complete sequencing data are available at GSE276849—GEO Publications (nih.gov).

### 4.1. Cell Culture Works

The human *KRAS*-mutated colorectal adenocarcinoma cell line HCT-15 and human duodenal adenocarcinoma HuTu80 cell line were kindly provided by Dr. Boris Margulis and Dr. Irina Guzhova from the Institute of Cytology, Russian Academy of Sciences, St. Petersburg. The HCT-15 cells were cultured in RPMI medium, and the HuTu80 cells were cultured in DMEM-F12 medium. Both media were supplemented with 10% fetal bovine serum (Biosera, Nuaille, France), 2 mM L-glutamine, and a penicillin–streptomycin mixture at a final concentration of 1% (Paneco, Russia). The cells were incubated in flasks or well plates at 37 °C in a humidified atmosphere containing 5% CO_2_. Once the cultures reached confluency, sub-culturing was performed using Versen solution (Paneco, Moscow, Russia) at a subculture ratio of 1:6, conducted every 3 days.

### 4.2. Generation of Knockout Mutant Cells

To generate mutant cells with knockout of the genes of interest, we first established homogeneous cell lines using the limiting dilution cloning method to create monoclonal cultures. Single cells were plated in 96-well plates (8 cells per mL), expanded to 24-well plates, and subsequently transferred to 6-well plates and T-25 flasks for further growth.

CRISPR/Cas9-mediated biallelic mutation of the *SLC39A14* gene was conducted in both the HCT-15 and HuTu80 cell lines, while knockout of the *SLC39A8* gene was specifically performed in HuTu80 cells. For guide RNA (gRNA) selection, we utilized the sequences provided by the Human CRISPR Knockout Pooled Library and the Crispick service. Guides were selected to target regions proximal to the start codon of the coding sequence, thereby facilitating gene knockout through the introduction of a frameshift mutation at both alleles. We designed three vectors for each gene, as the efficiency of individual gRNAs can vary significantly.

To express gRNAs targeting *SLC39A14* or *SLC39A8,* the following oligonucleotide duplexes were cloned into the pSpCas9(BB)-2A-GFP (PX458) vector (Addgene plasmid #48138; http://n2t.net/addgene:48138, accessed on 20 October 2021; RRID:Addgene_48138) [50]. 

-*SLC39A14*: forward sequence: 5′-CACCGCTAATACATCGGTATGGCGA-3′; reverse sequence: 5′-AAACTCGCCATACCGATGTATTAGC-3′.

-*SLC39A8*: forward sequence: 5′-CACCGTGCTTGGGCCGATCCTCACA-3′; reverse sequence: 5′-AAACTGTGAGGATCGGCCCAAGCAC-3′.

The HCT-15 and HuTu80 cell lines were transfected with the gRNA-Cas9 constructs using Lipofectamine 3000 (ThermoFisher, MA, USA). Following transfection, cells expressing GFP were enriched through fluorescence selection. GFP-positive cells were then sorted using a CytoFLEX SRT (Beckman Coulter, CA, USA) into 12-well plates. After reaching confluency, the cells were sorted to 96-well plates using limiting dilution cloning to establish isogenic clones.

To verify the presence of mutations at the targeted loci, genomic DNA was extracted from the HCT-15 and HuTu80 cells using the DNeasy Blood & Tissue Kit (QIAGEN). The region surrounding the gRNA target sites was PCR amplified, and the resulting products were subjected to cloning and Sanger sequencing. Sequence data were analyzed using SnapGene 1.1.3 software, and mutations were confirmed with TIDE, an automated tool for sequence trace decomposition [51]. Isogenic clones confirmed to harbor mutations were subsequently RNA sequenced.

### 4.3. Generation of the Stable Cell Lines with Overexpression of Genes of Interest

The pCDH-EF1-MCS-IRES-copGFP plasmid (SBI, USA) in which the GFP was replaced by the puromycin resistance selection marker and the *SLC39A8* (NM_022154.5), *SLC30A3* (NM_003459.5), SLC30A10 (NM_018713.3), and *SLC39A14* (NM_015359.6) ORF clones (Origene) were utilized for lentiviral vector construction. The genes *SLC39A14*, *SLC39A8*, *SLC30A3*, and *SLC30A10* were amplified using high-fidelity DNA polymerase with primers designed to include EcoRI and NotI restriction sites. PCR amplification was performed under the following conditions: an initial denaturation at 95 °C for 5 min, followed by 30 cycles of 95 °C for 30 s, an annealing step for 30 s, and an extension at 72 °C for 1 min, concluding with a final extension at 72 °C for 10 min. The PCR products were purified, digested by EcoRI and NotI (Thermo Fisher, MA, USA), and further purified by agarose gel electrophoresis. The digested inserts were then ligated into the pCDH-GFP vector using T4 DNA ligase and transformed into DH5α E. coli. The resulting colonies were screened by PCR.

HEK293T cells were then co-transfected with the recombinant plasmid along with packaging plasmids (psPAX2 and pMD2.G) using Lipofectamine 2000 (ThermoFisher, MA, USA) according to the manufacturer’s instructions. Viral supernatants were harvested 48–72 h post-transfection. Target cells (HCT-15 and HuTu80) were transduced with the viral supernatants in the presence of polybrene (5 µg/mL) for 24–48 h. Following transduction, the cells were selected with puromycin (6.6 µg/mL) for a duration of 5–10 days, with media replacement every 2–3 days. Clones confirmed to harbor the transgene were RNA sequenced.

### 4.4. Western Blot Analysis

Cells were washed with PBS, and proteins were extracted using a standard RIPPA buffer (Tris-HCl pH 8, 50 mM, NaCl 150 mM, Triton X-100 1%, DOC 0.5%, SDS 0.1%) on ice and sonication for 30 s to extract the proteins. The protein concentration was determined with a BCA assay kit (Thermo Fisher Scientific, Inc., Waltham, MA, USA), and SDS-PAGE was conducted according to the manufacturer’s guidelines (Bio-Rad Laboratories, Inc., Hercules, CA, USA). For immunoblotting, primary antibodies against FLAG and GAPDH (Gentech, South San Francisco, CA, USA) were used at a dilution of 1:10,000, along with appropriate secondary antibodies. The detection of protein bands was achieved by Pierce ECL Western Blotting Substrate (Thermo Fisher Scientific, Inc., Waltham, MA, USA).

### 4.5. Fluorescence Microscopy

For staining, FluoZin-3 AM was loaded into cells following the manufacturer’s instructions as described in [52]. Briefly, a 2 mM stock solution of FluoZin-3 AM was prepared by dissolving 100 µg of FluoZin-3 AM in an appropriate volume of DMSO. This stock solution was subsequently diluted to a final concentration of 2 μM in Opti-MEM (Thermo Fisher Scientific, Waltham, MA, USA). Prior to dye loading, the cells were washed with phosphate-free HHBSS buffer. They were then incubated with dye-free Opti-MEM containing 3 μM FluoZin-3 AM and 2 μM Hoechst 33258 nuclear stain for 1 h at 37 °C in a 5% CO₂ atmosphere. Following incubation, the cells were washed again with HHBSS buffer. Imaging was conducted immediately after the final wash.

Immunofluorescent staining images of cultured cells were captured using the EVOSM7000 imaging system (Thermo Fisher Scientific, Waltham, MA, USA) with a 20× objective. The imaging employed specific filter sets: DAPI (357/447 nm) for nuclei stained with Hoechst 33342 (Servicebio, Wuhan Servicebio Technology Co., Ltd., Wuhan, China) and GFP (470/525 nm) for FluoZin. The acquired images were processed using ZEISS ZEN 3.2 (blue edition) software.

### 4.6. Measuring Cell Growth

The doubling rate of cells was determined by modeling exponential growth using GraphPad PRISM version 8.0.1 software as described in [53]. Briefly, cells were seeded at a density of 16,000 cells per well in 96-well plates. Starting the following day, measurements of the Relative Fluorescent Units (RFU) of the cells were captured every 24 h. Cell numbers were quantified in RFU using the fluorescent nucleic acid stain SYTO 62 (Thermo Fisher Scientific, Waltham, MA, USA) [54]. Prior to staining, the cells were washed with phosphate-buffered saline (PBS) and permeabilized with 70% ethanol. After another wash with PBS, the cells were incubated with media containing 3 μM SYTO nuclear stain for 30 min at 37 °C in a 5% CO₂ atmosphere. Signal detection was conducted immediately after the final wash on dry cells in 96-well plates using a Varioskan Microplate Reader (Thermo Scientific, Waltham, MA, USA). The fluorescence intensity versus experimental time dependence was plotted, and the exponential growth equation was applied for the quantitative analysis of cell proliferation.

### 4.7. Resazurin-Based Cellular Respiration Assay

Cells were seeded into 96-well culture plates at a density of approximately 12,000 cells per well in 0.15 mL of growth medium. The plates were incubated for 24 h to allow for cell attachment. Following this incubation period, RFU measurements were obtained from live cells every 24 h, prior to SYTO staining. For the RFU measurement, the growth medium was removed, and a resazurin solution in PBS (0.1 mL per well) was added to each well. The plates were then incubated in the dark at 37 °C for 3 h to facilitate the accumulation of the resazurin reduction product—resorufin. The fluorescence intensity of resorufin was measured using a Varioskan Flash microplate reader (Thermo Scientific, Waltham, MA, USA), with an excitation wavelength of 550 nm and an emission wavelength of 590 nm. The RFU values obtained for raw mitochondrial respiration for each cell line were normalized to derive the normalized mitochondrial respiration coefficient by correlating the resorufin signal with the SYTO signal.

### 4.8. Measurements of Half-Inhibitory Concentrations (IC50) of Drugs

Cisplatin (Teva, Amsterdam, The Netherlands) was a 1.66 mM solution stored at room temperature.

The cell viability for IC50 calculations was assessed using the MTT Cell Proliferation Kit (Roche, Basel, Switzerland) according to the manufacturer’s protocol. Cells were cultured in 96-well plates and incubated with a 0.5 µg/mL solution of yellow MTT in the culture media for 3–4 h for the formation of purple formazan salt crystals. A solubilization buffer was then added, and the plates were incubated overnight in a humidified atmosphere at 37 °C with 5% CO_2_. The solubilized formazan product was quantified using a Varioskan Microplate Reader spectrophotometer, where the increase in absorbance directly correlated with the number of living cells and their metabolic activity.

IC50 values were calculated using GraphPad PRISM (version 8.0.1) software as described in [55]. The data were normalized to controls (untreated cells designated as 100% viability) and plotted as a dose–response curve. A four-parameter logistic model was employed to fit the data, facilitating the determination of the drug concentration required to inhibit cell viability by 50% (IC50). All experiments were conducted in triplicate, and the results are presented as the mean ± standard deviation.

### 4.9. Cell Cycle Analysis

Cell cycle analysis was performed according to [56]. The cells were seeded in 6-well culture plates at a density of 50,000 cells per well and incubated for 3 days. The cells were detached using Versen solution. The detached cells were washed twice with PBS by centrifugation at 100× *g*. The resulting cell pellet was fixed in 70% ethanol at −20 °C for 30 min. Subsequently, the cells were centrifuged for 10 min at 3800 rpm (1000× *g*), washed with PBS, and resuspended in 0.2 mL of PBS containing 10 μg/mL RNase A II and 2 μg/mL propidium iodide. The samples were incubated at room temperature for 30 min in the dark. Flow cytometric analysis was performed using a BD Accuri C6 Plus instrument to assess cell cycle distribution.

### 4.10. Preparation of Libraries and RNA Sequencing

RNA sequencing libraries were prepared and sequenced as described in [57]. Total RNA was isolated using the RecoverAll Total Nucleic Acid Isolation Kit (Invitrogen, Waltham, MA, USA). The concentration of RNA was quantified using a Qubit RNA Assay Kit, while the RNA Integrity Number (RIN) was assessed with an Agilent 2100 Bioanalyzer. Ribosomal RNA was depleted using the RNA Hyper Kit from Roche. Subsequently, the library concentrations and fragment size distributions were analyzed using Qubit (Thermo Fisher Scientific, Waltham, MA, USA) and Agilent TapeStation (Agilent Technologies, Santa Clara, CA, USA), respectively. RNA sequencing was performed on an Illumina NextSeq550 engine, generating 50 bp single-end reads, with each sample yielding approximately 30 million raw reads.

### 4.11. Processing of RNA Sequencing Data

Raw sequencing reads were quality assessed using the Illumina Sequencing Analysis Viewer and subsequently demultiplexed with Illumina Bcl2fastq2 version 2.17 software. The resulting FASTQ files were processed and aligned using the STAR program in “GeneCounts” mode, utilizing the Ensembl human transcriptome annotation (version GRCh38.89). Ensembl gene identifiers were converted to Human Gene Nomenclature Committee (HGNC) gene symbols via the complete HGNC dataset (version 13 July 2017), enabling expression level determination for 36,596 annotated genes with HGNC identifiers.

For literature expression datasets, RNA sequencing profiles were sourced from the Cancer Genome Atlas (TCGA) project using the TCGAbiolinks package. Additionally, a proprietary database containing around 1300 original RNA sequencing profiles from human tumor specimens was utilized, which included 154 profiles of colorectal cancer (RTC), with 25 specimens confirmed to have activating *KRAS* mutations. Molecular pathway information was extracted from the OncoboxPD Web database [47].

Pathway activation levels (PALs) were calculated using the Oncobox method [16], employing the natural logarithm for calculations. The *p*-value was defined as a quantile of PALs in the investigated sample relative to the PAL distribution in normal samples. Both positive and negative PAL values were considered potential prognostic biomarkers for colorectal cancer.

Logarithmically transformed read counts, normalized using DESeq2 software (version 1.44.0), facilitated hierarchical clustering analysis of transcriptome profiles. Genes deemed differentially expressed were required to meet a threshold of Benjamini–Hochberg false discovery rate (FDR)-adjusted *p*-value < 0.05. Volcano plots were generated using the R package EnhancedVolcano (version 1.16.0), and statistical analyses were conducted in R. Gene Ontology (GO) enrichment analysis was performed with the R packages clusterProfile (version 4.2.1) and org.Hs.eg.db (version 3.8.2). Principal Component Analysis (PCA) and the visualization of log-transformed counts for all genes were executed using pca2d R (version 3.6.2) and prcomp (from “stats” package version 4.4.0) software. Venn diagram plotting was performed using the R packages ggVennDiagram (version 1.5.2) and RVenn (version 1.1.0).

### 4.12. Statistical Analysis

Statistical analysis was performed using GraphPad PRISM 8.0.1 software. A *p*-value of less than 0.05 was considered statistically significant, with results presented as follows: *p*  >  0.05 (non-significant, ns), *p*  ≤  0.05 (*), *p*  ≤  0.01 (**), *p*  ≤  0.001 (***), and *p*  ≤  0.0001 (****). Data from two groups were analyzed using an unpaired *t*-test, while data involving more than two groups were assessed using either one-way or two-way ANOVA, followed by Tukey’s post hoc analysis to determine specific group differences. The line-of-best fit was determined using GraphPad Prism, and the slopes of the lines were compared through linear regression analysis. Data are presented as the mean ± standard deviation (SD) of at least three independent experiments conducted on different days.

The significance of pathway activation levels (PALs) was evaluated using the Oncobox pathway analysis method [16] with OncoboxPD (Oncobox pathway databank) [47]. For PAL calculations, each expression profile was normalized by the mean geometric expression level of all genes within the dataset under investigation.

## 5. Conclusions

In this study, we explored the roles of the SLC30A and SLC39A gene families in modulating cancer cell behavior, focusing on zinc and manganese transport in human colorectal and duodenal carcinoma cell lines. Our findings provide novel insights into the impact of altered expression of these genes on cell viability, mitochondrial function, and drug sensitivity, particularly to cisplatin. The role of metal ion transport proteins, particularly ZnT10 encoded by the *SLC30A10* gene, has been previously examined in connection with carcinogenesis, specifically in CRC. In this study, we demonstrated significant cisplatin resistance in HCT-15 and HuTu80 cells with overexpression of the *SLC30A10* gene, which is involved in antioxidant defense mechanisms, as zinc serves as a cofactor for enzymes such as superoxide dismutase. Further research will help to uncover the precise reason for the resistance to platinum-based chemotherapy associated with mutations in the ion channel gene—whether it is due to competition between ions for cellular uptake or if the resistance is provoked by an enhanced ability of mitochondria to cope with oxidative stress as a result of *SLC30A10* overexpression.

Overall, our findings suggest that the interplay between the SLC30A and SLC39A gene families may play a significant role in the regulation of cancer cell sensitivity to platinum-based chemotherapy—cisplatin. These results highlight the potential of using the expression status of these genes as biomarkers for predicting tumor responsiveness to treatment, which warrants further investigation in future studies.

## Figures and Tables

**Figure 1 ijms-25-12049-f001:**
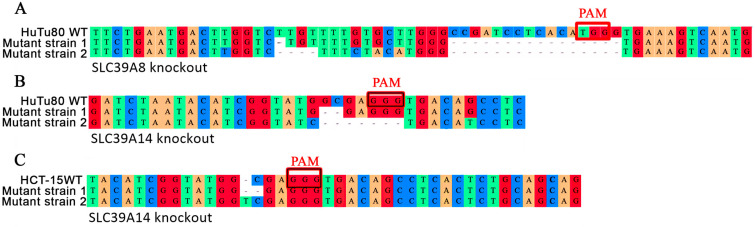
Aligned decomposited sequences of mutant clones versus the sequences of wild-type cells. (**A**) HuTu80 knockout by targeting exon 3 of *SLC39A8* gene; (**B**) HuTu80 cell knockout by targeting exon 2 of *SLC39A14* gene; (**C**) HCT-15 cell knockout by targeting exon 2 of *SLC39A14* gene.

**Figure 2 ijms-25-12049-f002:**
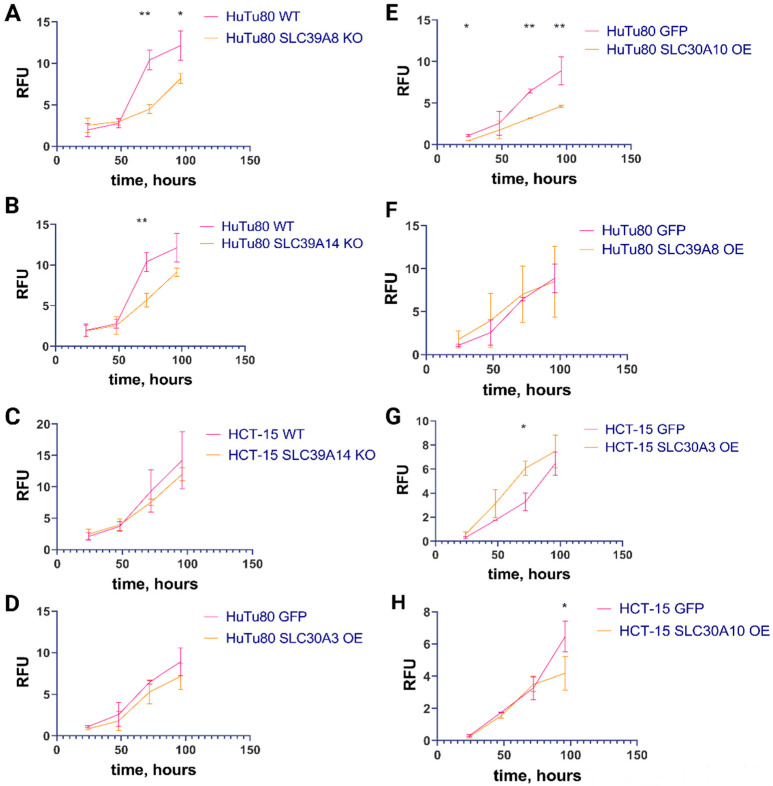
Growth curves indicating the growth rates for parental and mutant cell lines during 4 days of daily measurements. (**A**) Comparison of HuTu80 WT cell line and HuTu80 cell line with knockout of *SLC39A8* gene. Significant difference in cell amount (measured in Relative Fluorescent Units, RFU) at 72 h and 96 h of incubation. (**B**) Comparison of HuTu80 WT cell line and HuTu80 cell line with knockout of *SLC39A14* gene. Significant difference in cell amount (measured in RFU) at 72 h of incubation. (**C**) Comparison of HCT-15 WT cell line and HCT-15 cell line with knockout of *SLC39A14* gene. (**D**) Comparison of HuTu80 GFP control cell line and HuTu80 cell line with overexpression of *SLC30A3* gene. (**E**) Comparison of HuTu80 GFP control cell line and HuTu80 cell line with overexpression of *SLC30A10* gene. Significant difference in cell amount (measured in RFU) at 24 h, 72 h, and 96 h of incubation. (**F**) Comparison of HuTu80 GFP control cell line and HuTu80 cell line with overexpression of *SLC39A8* gene. (**G**) Comparison of HCT-15 GFP control cell line and HCT-15 cell line with overexpression of *SLC30A3* gene. Significant difference in cell amount (measured in RFU) at 72 h of incubation. (**H**) Comparison of HCT-15 GFP control cell line and HCT-15 cell line with overexpression of *SLC30A10* gene. Cell number was quantified in RFU using the fluorescent nucleic acid stain SYTO 62. Data (mean ± SEM) from three or more independent experiments are presented. *, *p* < 0.05; **, *p* < 0.01.

**Figure 3 ijms-25-12049-f003:**
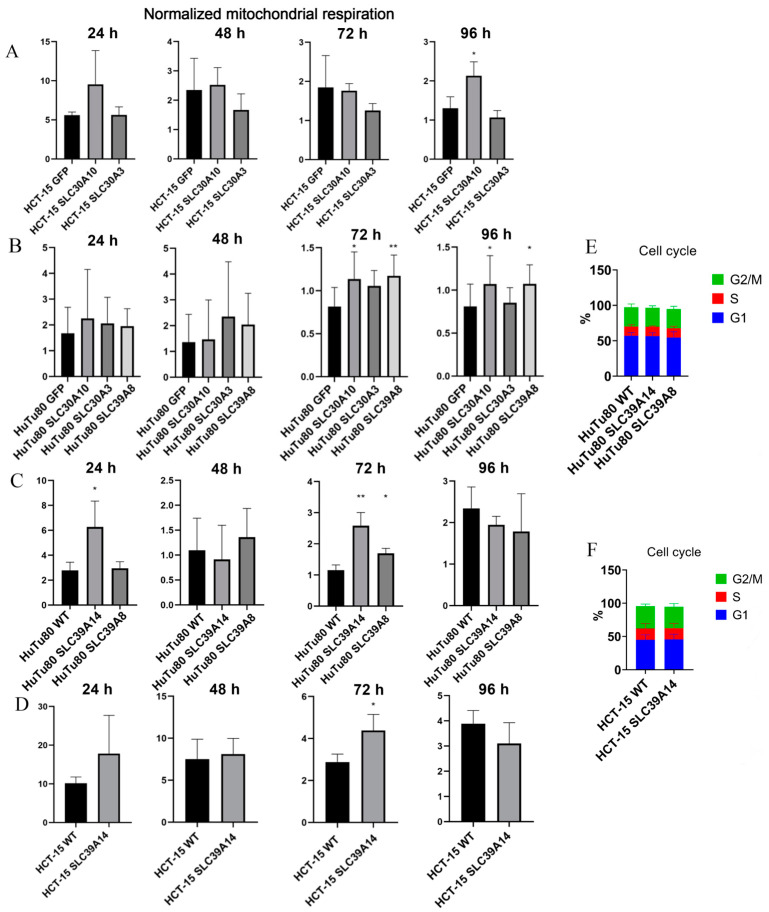
Resazurin assay metabolism measurements for parental and mutant cell lines during 4 days of incubation and cell cycle analysis of knockout mutant cells. (**A**) HCT-15 GFP and cells with overexpression; (**B**) HuTu80 GFP and cells with overexpression; (**C**) HuTu80 parental and knockout mutant cells; (**D**) HCT-15 parental and knockout mutant cells; (**E**) cell cycle analysis of knockout and parental HuTu80 cells in standard conditions; (**F)** cell cycle analysis of knockout and parental HCT-15 cells in standard conditions. Resorufin fluorescence normalized per RFU of nuclei staining with SYTO 62. Data (mean ± SEM) from three or more independent experiments are presented. *, *p* < 0.05; **, *p* < 0.01.

**Figure 4 ijms-25-12049-f004:**
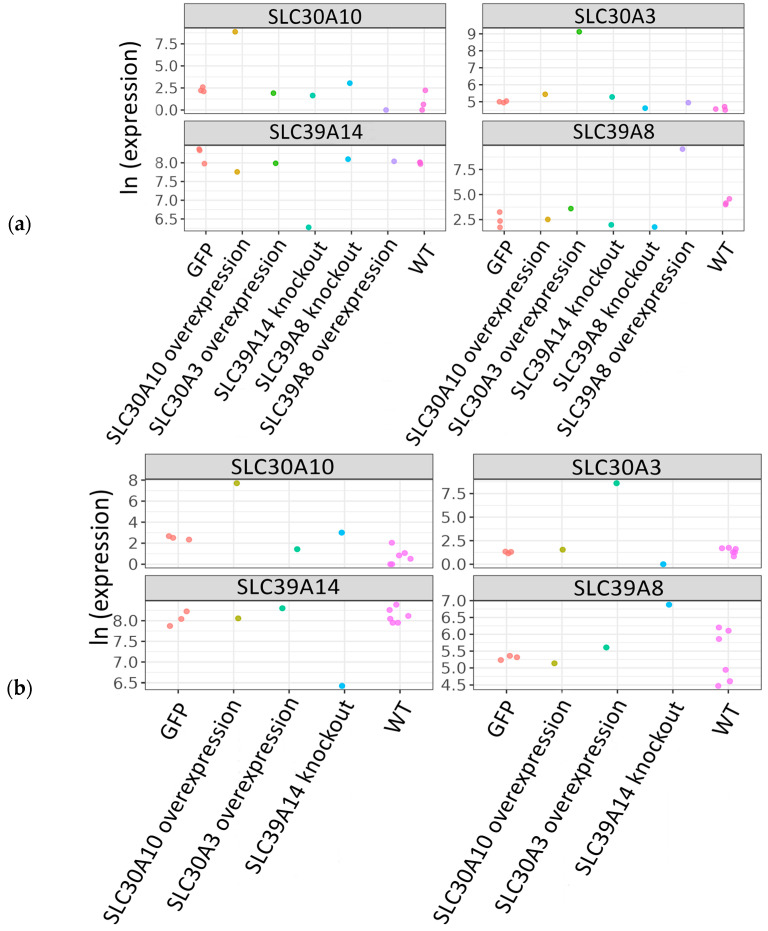
Box plots depicting the expression levels of SLC genes in both parental and mutant cell lines are presented. The color key indicates functional groups of samples. (**a**) The differential expression analysis compares knockout clones to WT controls, as well as clones with overexpression against HuTu80 cells containing a GFP-vector control. (**b**) The differential expression analysis compares knockout clones to WT controls, in addition to clones with overexpression against HCT-15 cells with a GFP-vector control.

**Figure 5 ijms-25-12049-f005:**
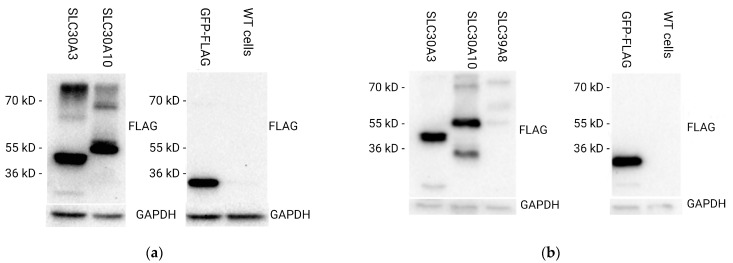
Western blot analysis of cells transduced by lentiviral vector: (**a**) HCT-15 cells transduced by *SLC30A3* and *SLC30A10* constructions. Control HCT-15 cells with GFP-FLAG and WT cells; (**b**) HuTu80 cells transduced by *SLC30A3*, *SLC30A10*, and *SLC39A8* constructions. Control HuTu80 cells with GFP-FLAG and WT cells. GAPDH was used as a loading control.

**Figure 6 ijms-25-12049-f006:**
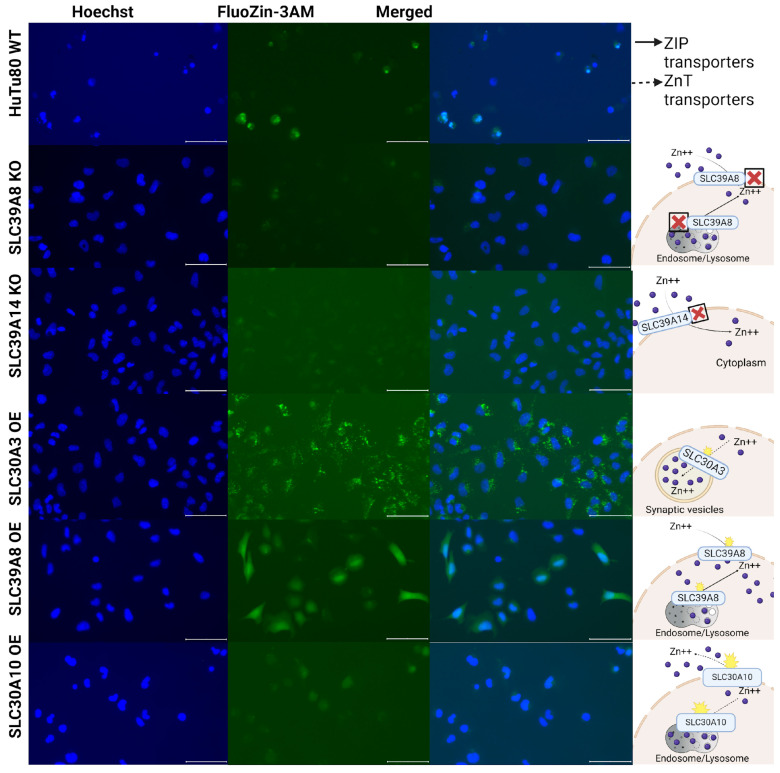
Measurement of free zinc levels in the mutant and parental HuTu80 cell lines. HuTu80 cells were stained with FluoZin-3 AM and Hoechst 33342. Scale bar, 100 μm.

**Figure 7 ijms-25-12049-f007:**
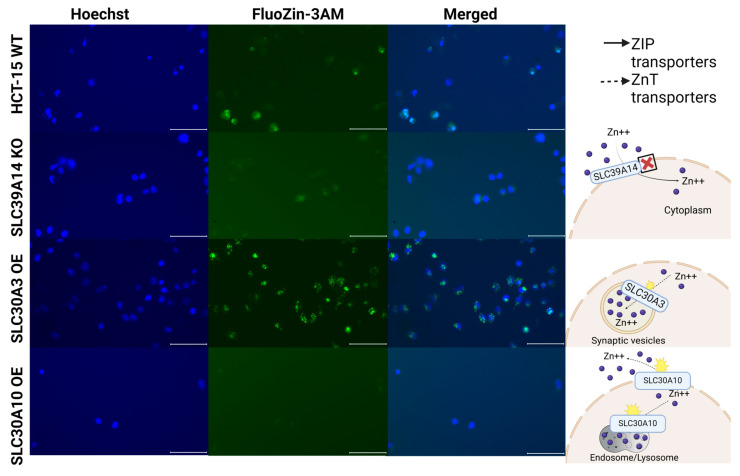
Measurement of free zinc levels in the mutant and parental HCT-15 cell lines. HCT-15 cells were stained with FluoZin-3 AM and Hoechst 33342. Scale bar, 100 μm.

**Figure 8 ijms-25-12049-f008:**
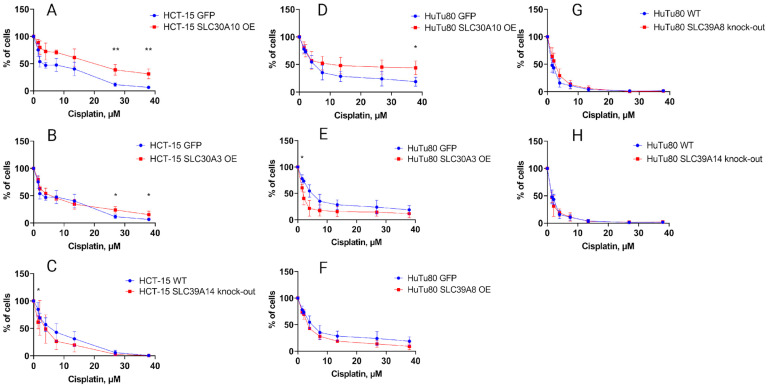
Cell proliferation inhibition by cisplatin. (**A**) HCT-15 with *SLC30A10* overexpression vs. HCT-15 GFP control; (**B**) HCT-15 with *SLC30A3* overexpression vs. HCT-15 GFP control; (**C**) HCT-15 cells with *SLC39A14* knockout vs. HCT-15 WT control; (**D**) HuTu80 cells with *SLC30A10* overexpression vs. HuTu80 GFP control; (**E**) HuTu80 cells with *SLC30A3* overexpression vs. HuTu80 GFP control; (**F**) HuTu80 cells with *SLC39A8* overexpression vs. HuTu80 GFP control; (**G**) HuTu80 cells with *SLC39A8* knockout vs. HuTu80 WT control; (**H**) HuTu80 cells with *SLC39A14* knockout vs. HuTu80 WT control. Cells were grown in 96-well plates and treated with various concentrations of drug (cisplatin: 0–37 μmol/L)**.** *, *p* < 0.05; **, *p* < 0.01.

**Figure 9 ijms-25-12049-f009:**
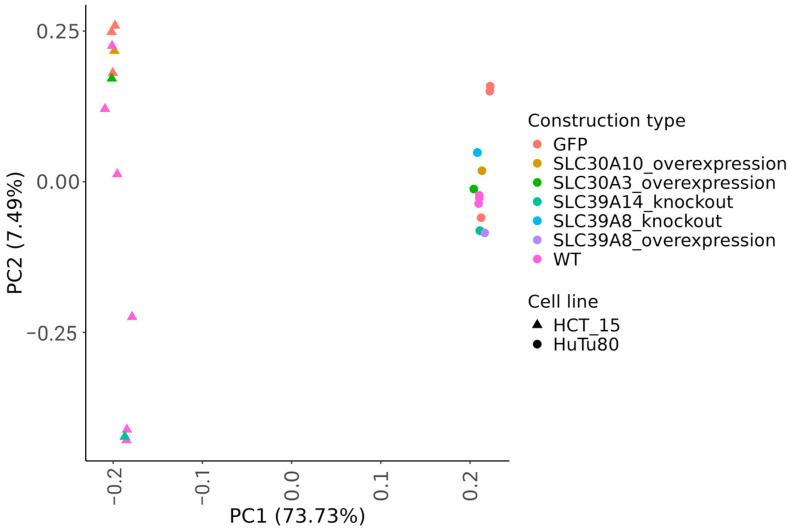
Principal component analysis (PCA) of log-transformed gene expression profiles. The analysis compares knockout clones to WT controls, as well as clones with overexpression against cells with a GFP-vector control. The color key indicates functional groups of samples. The proportion of standard deviation for the respective PC is shown in brackets.

**Table 1 ijms-25-12049-t001:** Cisplatin IC50 values for HuTu80 and HCT-15 intact and mutant cells. Data (mean ± SEM) from three or more independent experiments are presented. Mean differences compared with control cells. ** *p* < 0.01; *** *p* < 0.001.

Cell Line	Cisplatin IC50, µM	*p*-Value
HuTu80 WT	1.52	
HuTu80 *SLC39A8* KO	2.3	***
HuTu80 *SLC39A14* KO	1.3	n/a
HuTu80 GFP	5.3	
HuTu80 *SLC30A3* OE	1.6	***
HuTu80 *SLC30A10* OE	14.5	***
HuTu80 *SLC39A8* OE	3.6	**
HCT-15 WT	5	
HCT-15 *SLC39A14* KO	2.9	**
HCT-15 GFP	4.3	
HCT-15 *SLC30A3* OE	4.8	n/a
HCT-15 *SLC30A10*	17.35	***

## Data Availability

The normalized RNA sequencing counts are available in GEO with the accession number GSE276849.

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
