# Peer review of "Forced Overexpression and Knockout Analysis of SLC30A and SLC39A Family Genes Suggests Their Involvement in Establishing Resistance to Cisplatin in Human Cancer Cells"

_ijms, 2024, doi:10.3390/ijms252212049_

Round 1
Reviewer 1 Report
Comments and Suggestions for Authors
The authors investigated the effects of overexpression or knockout of SLC30A family and SLC39A family genes on the cisplatin sensitivity of human cancer cells, such as HCT-15 and HuTu80. This is a meaningful study with promising potential to identify new cancer therapeutic targets. However, there are still some issues that need to be addressed:
1. Figure 3E, F,Can you use different colors for G2/M and S phases of the cell cycle? It's hard to distinguish them in the current graph.
2. In Figures 6 and 7, the localization of FluoZin-3 is not clear enough. Please provide clearer images showing the localization of FluoZin-3 in the cytoplasm or endosome/lysosome. Alternatively, you could use endosome/lysosome staining dyes to co-localize with FluoZin-3, making the results clearer.
3. In Figure 8A, B, is there a statistical difference in cell proliferation inhibition between different groups (e.g., GFP, SLC30A10 OE, SLC30A3 OE, WT, etc.)? Please indicate the statistical significance in the figure.
Reviewer 2 Report
Comments and Suggestions for Authors
The current manuscript entitled ‘Forced overexpression and knockout analysis of SLC30A and SLC39A family genes suggests their involvement in establishing resistance to cisplatin in human cancer cells’ provides insights into the roles of SLC30A and SLC39A gene families in regulating zinc and manganese transport and their impact on colon cancer cell survival and chemotherapy sensitivity, particularly to cisplatin. The use of genetically modified cancer cell lines, HCT-15 and HuTu80, enables a detailed investigation into how these genes influence cancer pathways. A key finding is that SLC30A10 overexpression increases cisplatin resistance, while SLC30A3 overexpression enhances sensitivity, highlighting their potential as biomarkers for platinum-based therapies.
However, several aspects can be improved or require clarification from the authors. The detailed comments are as follows:
1. Please include more background information on the role of mitochondrial respiration in cancer development in the Introduction Section.
2. According to Figure 2, gene knockout and overexpression do affect the growth rate of cells, please clarify its potential effect on the study results. In addition, please indicate the doubling times before and after gene modulation for each cell line.
3. You indicated that after gene knockout, the protein expression was reduced. Please confirm it by western blot. In addition, although the protein level is reduced, can you confirm if these proteins still have function or not (both enzymatic and scaffold functions)?
4. Regarding the gene overexpression, why not perform the SLC39A14 overexpression? In addition, please point out the target proteins in Figure 5, and why there are no target proteins in the WT cells.
5. For Figure 8, please make a table for IC50s and show each IC50 curve independently.
6. Please add heatmap figures regarding the gene up and down regulations.
7. The current discussion section is not comprehensive, the resulted data are not sufficiently discussed. Please provide a more detailed discussion, e.g., SLC39A8 overexpression increases the IC50 of cisplatin, however, SLC39A8 knockout does not affect the cell sensitivity to cisplatin treatment. What’s the potential relationship between mitochondrial respiration changes and cell sensitivity to cisplatin treatment?
8. Please include a conclusion section in the current manuscript.
Reviewer 3 Report
Comments and Suggestions for Authors
The authors explored the SLC30A/SLC39A gene family in the context of cancer drug resistance, suggesting they could be potential targets for colon cancer. However, the manuscript's presentation is chaotic and disorganized. The figures lack clarity, and there is no in vivo study to support their conclusions. I believe the manuscript requires significant reorganization.
Concerns:
- In Figure 2, please label the siSLC30A and overexpression conditions clearly (e.g., "OE").
- Provide evidence of the knockout effect using Western blot analysis.
- Typically, two knockout sequences should be employed to ensure there are no off-target effects.
- The title should be shortened and refined.
- Ensure uniformity in font labels, including size and style.
- Review the manuscript for grammatical errors and typos.
NONE
Round 2
Reviewer 2 Report
Comments and Suggestions for Authors
In the current response letter, the authors have provided clear and satisfactory responses to the reviewers’ comments and have made the corresponding revisions to the manuscript. After reviewing the updated version, the overall recommendation is 'Accept in present form'.
Author Response
Comments: In the current response letter, the authors have provided clear and satisfactory responses to the reviewers’ comments and have made the corresponding revisions to the manuscript. After reviewing the updated version, the overall recommendation is 'Accept in present form'.
Response: We would like to express our sincere gratitude for Reviewer's thorough and insightful review of our manuscript titled “Forced overexpression and knockout analysis of SLC30A and SLC39A family genes suggests their involvement in establishing resistance to cisplatin in human cancer cells”. Reviewer's expertise and constructive feedback have significantly contributed to the improvement of our work.
Reviewer 3 Report
Comments and Suggestions for Authors
Thanks for the author addressing my concerns, from my perspective, I have no further questions. Please make the figures clear with high resolutions.
Author Response
Comments: Thanks for the author addressing my concerns, from my perspective, I have no further questions. Please make the figures clear with high resolutions.
Response: We sincerely appreciate the time and effort you dedicated to evaluating our research. Your comments prompted us to clarify several key points and enhance the overall quality of the manuscript. In response to your feedback regarding the clarity of the figures, we have resubmitted new versions of all figures with significantly improved resolution, about 3000 pixels in vector graphics. We have paid special attention to Figure 2 (Results, page 5, line 160), and we hope that the new version ensures both clarity and detail.